# A high infectious simian adenovirus type 23 vector based vaccine efficiently protects common marmosets against Zika virus infection

**Shengxue Luo[1]☯, Wei Zhao[2]☯, Xiaorui Ma[1], Panli Zhang[1], Bochao Liu[1], Ling Zhang[1], Wenjing Wang[1], Yuanzhan Wang[3], Yongshui Fu[4], Jean-Pierre Allain[1,5], Tingting Li[1]\*, Chengyao Li [1]\***

**1** Department of Transfusion Medicine, School of Laboratory Medicine and Biotechnology, Southern Medical University, Guangzhou, China, **2** Laboratory of Biosafety, School of Public Health, Southern Medical University, Guangzhou, China, **3** Experimental Animal Center, Nanfang Hospital, Southern Medical University, Guangzhou, China, **4** Guangzhou Blood Center, Guangzhou, China, **5** Emeritus professor, University of Cambridge, Cambridge, United Kingdom

☯ These authors contributed equally to this work.
\* apple-ting-007@163.com (TL); chengyaoli@hotmail.com (CL)

**Data Availability Statement:** All relevant data are within the manuscript and its Supporting Information files.

## Abstract

Zika virus (ZIKV) has spread in many countries or territories causing severe neurologic complications with potential fatal outcomes. The small primate common marmosets are susceptible to ZIKV, mimicking key features of human infection. Here, a novel simian adenovirus type 23 vector-based vaccine expressing ZIKV pre-membrane-envelope proteins (Sad23L-prM-E) was produced in high infectious titer. Due to determination of immunogenicity in mice, a single-dose of $3\times10^8$ PFU Sad23L-prM-E vaccine was intramuscularly inoculated to marmosets. This vaccine raised antibody titers of $10^{4.07}$ E-specific and $10^{3.13}$ neutralizing antibody (NAb), as well as robust specific IFN-γ secreting T-cell response (1,219 SFCs/$10^6$ cells) to E peptides. The vaccinated marmosets, upon challenge with a high dose of ZIKV ($10^5$ PFU) six weeks post prime immunization, reduced viremia by more than 100 folds, and the low level of detectable viral RNA ($<10^3$ copies/ml) in blood, saliva, urine and feces was promptly eliminated when the secondary NAb (titer $>10^{3.66}$) and T-cell response ($>726$ SFCs/$10^6$ PBMCs) were acquired 1–2 weeks post exposure to ZIKV, while non-vaccinated control marmosets developed long-term high titer of ZIKV ($10^{5.73}$ copies/ml) ($P<0.05$). No significant pathological lesions were observed in marmoset tissues. Sad23L-prM-E vaccine was detectable in spleen, liver and PBMCs at least 4 months post challenge. In conclusion, a prime immunization with Sad23L-prM-E vaccine was able to protect marmosets against ZIKV infection when exposed to a high dose of ZIKV. This Sad23L-prM-E vaccine is a promising vaccine candidate for prevention of ZIKV infection in humans.

**Funding:** This work was supported by the funding from the National Natural Science Foundation of China (No. 31770185, 31970886, 81871655 and 31500134) for CL and TL, the National Key Research and Development Program (No. 2017YFD0500305) for CL, the Guangdong Province Universities and Colleges Pearl River Scholar Funded Scheme (2017) for TL, the Guangzhou major project of industry-university-research cooperation and collaborative innovation (No. 201704020083) for TL, and the Innovative R&D Team Introduction Program of Guangdong (No. 2014ZT05S123) for CL. The funders had no role in study design, data collection and analysis, decision to publish, or preparation of the manuscript.

**Competing interests:** The authors have declared that no competing interests exist.

## Author summary

Zika virus (ZIKV) is a member of the *Flaviviridae* family) and causes severe neurologic diseases. The development of safe and effective vaccine is urgent need. In this study, we constructed a novel simian adenovirus type 23 vector-based vaccine expressing ZIKV pre-membrane-envelope proteins (Sad23L-prM-E). By vaccinating the common marmosets with prime immunization of vaccine, and upon challenge with a high dose of ZIKV to the vaccinated marmosets, the immune response and protective efficacy of vaccine were extensively evaluated. The data suggested that Sad23L-prM-E vaccine could protect marmosets against a high dose of ZIKV challenge, which provided a promising vaccine for preventing ZIKV infection in humans.

## Introduction

Zika virus (ZIKV) was isolated from the blood of rhesus macaque in the Zika Forest of Uganda in 1947 [1]. Following an outbreak of ZIKV infection occurring in 2015 in Brazil [2], it became a global public health issue. ZIKV is mainly transmitted by mosquitoes, causing ZIKV-related congenital syndrome (microcephaly, brain calcifications, congenital central nervous system anomalies, stillbirths, hypertonia) [3–7], and is linked to Guillain–Barré syndrome in adults [8,9]. Currently, the development of safe and effective vaccine is in order.

ZIKV is a positive-sense single-stranded RNA virus in the *Flaviviridae* family). The E protein was identified as inducing neutralizing antibodies (NAb) to ZIKV infection [10,11], while the prM protein is integral part of both virion and sub-viral particle undergoing cleavage during virus maturation [4,5,12,13,14]. Therefore, prM and E proteins (prM-E) have been the primary targets in designing recombinant vaccines [15–18].

Simian adenoviruses have low-seroprevalence in humans [19,20]. It was previously reported that vaccination with recombinant rhesus adenovirus type 52 vector (RhAd52) or chimpanzee adenovirus type 7 vector (AdC7) expressing ZIKV prM-E induced protective immunity against ZIKV challenge in mice and rhesus macaques [21,22], but the infectious titer of vaccines was relatively low.

Common marmosets are New World small primates having the advantages of small size for easy handling, gentle disposition and a compressed lifespan compared to other non-human primates (NHPs), carrying immunological markers similar to those of humans, which make this small monkey highly attractive for biomedical studies [23,24]. Recently, marmosets were found susceptible to ZIKV, mimicking key features of human infection and fetal neurocellular disorganization [25,26]. Marmosets are therefore an attractive model for evaluation of ZIKV vaccine efficacy.

In this study, the novel simian adenoviral vector Sad23L constructed as a derivate of simian adenovirus type 23 (SAdV23) with high viral titer was used to develop a ZIKV vaccine for delivering ZIKV prM-E antigens. The immunogenicity and protective efficacy of Sad23L-prM-E vaccine were extensively evaluated in mice and marmosets.

## Materials and methods

### Ethics statement

The use of common marmosets (*Callithrix jacchus*) for this study was approved by the authorities of Forest Bureaus, Guangdong and Tianjin governments (Yuelinhu [2014]160), respectively. Ethical approval for the marmoset experimentation and sample collection was also

obtained from the Southern Medical University (SMU) Animal Care and Use Committee at Nanfang hospital, SMU, Guangzhou [permit numbers: SYXK (Yue) 2010–0056]. All animal care and experimental procedures (NFYYLASOP-037) were in accordance with national and institutional policies for animal health and well-being.

The welfare issues (housing, feeding, environmental enrichment *et al*) were in accordance with the recommendations of the Weatherall report (https://acmedsci.ac.uk/more/news/the-use-of-non-human-primates-in-research). The animals were individually housed in spacious cages and were provided with commercial food pellets supplemented with appropriate treats. Drinking water was provided at libitum from an automatic watering system. Enrichment was provided in the form of pieces of wood, a variety of food and other home made or commercially available enrichment products. Animals were monitored daily for health and discomfort. For all procedures, animals were anesthetized with an intramuscular dose of ketamine (10mL/kg). Blood samples were obtained using a vacutainer or needle and syringe from the femoral. Marmosets were challenged with Asian ZIKV Z16006.

## Viruses and cells

The simian adenovirus type 23 (SAdV23/AdC6/Pan6, ATCC-VR-592) was purchased from American Type Culture Collection (ATCC). ZIKV isolate of Asian lineage Z16006 strain (Gen-Bank no. KU955589.1) was isolated from a patient who travelled to Fiji and Samoa by the Center for Disease Control and Prevention of Guangdong Province, China in 2016, which was provided by Professor Wei Zhao (School of Public Health, Southern Medical University, China).

African green monkey kidney epithelial cells (Vero cells), HEK-293 cells and Huh7.5.1 were maintained in complete Dulbecco's modified Eagle's medium (DMEM; Gibco, NY, USA) supplemented with 10% fetal bovine serum (FBS; Gibco, NY, USA) and incubated in 5% $CO_2$ at 37˚C.

C57BL/6 mice were obtained from the Animal Experimental Centre of Southern Medical University, Guangdong, China.

## Production of Zika virus vaccine Sad23L-prM-E

A chimeric simian adenoviral vector Sad23L was constructed by deleting the E1 and E3 regions of the full-length simian adenovirus serotype 23 genome (SAdV23) [27]. The E4 region open reading frame 6 (orf6) was replaced by the corresponding element of human adenovirus type 5 (Ad5) in order to improve virus propagating efficiency. The Japanese encephalitis virus (JE) signal peptide gene (encoding for JE signal peptide MGKRSAGSIMWLASLAVVIA-CAGA) was synthesized commercially (Beijing Genomics Institute, Beijing, China) [28] and ZIKV pre-membrane-envelope (prM-E) genes from ZIKV-Z16006 strain were cloned into the deleted E1 region of Sad23L plasmid designated as Sad23L-prM-E. The recombinant adenovirus Sad23L-prM-E was rescued and propagated from HEK-293 cells and was serially passaged for 12 generations when full cytopathic effect appeared. The vaccines were purified by cesium chloride density gradient centrifugation as previously described [29].

## Western blotting

HEK-293, Vero, Huh7.5.1 and marmosets' peripheral blood mononuclear cells (PBMCs) were infected with Sad23L-prM-E virus, respectively, and Sad23L-empty virus was used as mock control. The expression of ZIKV E protein was analyzed by Western blotting with anti-ZIKV E antibody (BioFront Technologies, FL, USA). Glyceraldehyde-3-phosphate dehydrogenase (GADPH) was used as a loading control. The membranes were washed five times and

developed by Supersignal West Pico Plus chemiluminescent substrate (Thermo scientific, MA, USA).

## Immunofluorescence assay

Vero cells were infected with Sad23L-prM-E virus, and Sad23L-empty as mock control. Cells were fixed and incubated with anti-ZIKV E antibody, and extensively washed with PBST. Anti-mouse IgG-Alexa Fluor 594 antibody (Thermo scientific, MA, USA) in 1% BSA-PBST was added to the cells for 30 min at 37˚C. Diamidinophenylindoldiacetate (DAPI) was added to stain cell nuclei.

## Animal's immunization and challenge

Female C57BL/6 mice (5–6 weeks of age, n = 5 each group) were inoculated intramuscularly (i.m.) with $5\times10^6$, $5\times10^7$ and $5\times10^8$ PFU Sad23L-prM-E virus doses, $5\times10^8$ PFU Sad23L-empty viruses and an equivalent volume of PBS was used as sham controls.

Marmosets M34, M37, and M47 at 4 to 5 years of age (S1 Table) were immunized intramuscularly (i.m.) with $3\times10^8$ PFU Sad23L-prM-E (n = 3) according to the determination of immunogenicity in mice, the principle of "necessary and sufficient" and the weight ratio of marmoset to mouse. An equivalent volume of PBS was injected into M46 and M48 as sham controls (n = 2). Marmosets were then infected with $1\times10^5$ PFU of ZIKV-Z16006 viruses by the i.m. route at week 6 post vaccination.

## Enzyme-linked immunosorbent assay (ELISA)

The microtiter plates (Corning, NY, USA) were coated overnight with 5μg/ml of ZIKV E protein (Sino Biological, Beijing, China). Serum samples were 3-fold serially diluted and E binding antibody was detected by ELISA. Endpoint titers were defined as the highest reciprocal serum dilution giving an absorbance more than 2-fold (in mice) or over 5-fold (in marmosets) background values. Log10 end point titers were reported.

## ZIKV neutralization assay

Neutralizing antibody (NAb) titers were determined by a standard 50% plaque reduction neutralization test ($PRNT_{50}$) as previously described [30]. Briefly, serum was inactivated by heating at 56˚C for 30 min, then was 3-fold serially diluted and mixed with an equal volume of ZIKV-Z16006 (100 PFU). After incubation at 37˚C for 1h, an aliquot of 200μl serum-virus mixture was added to 24-well plate containing 90% confluent monolayers of Vero cells. After incubation in $CO_2$ at 37˚C for six days, the monolayers were fixed with formalin (10%) and stained with crystal violet (0.05%). The plaques were counted at a magnification of 12.5×. Endpoint titers were defined as the highest reciprocal serum dilution of 50% plaque reduction. Log10 $PRNT_{50}$ titer was reported.

## ELISpot

Human (or mouse) IFN-gamma ELISpotPLUS kits (MabTech, Sweden) were used to determine antigen-specific T lymphocyte response. Marmoset PBMCs ($2\times10^5$ cells/well) or mouse splenocytes ($5\times10^5$ cells/well) were stimulated with peptides (10μg/ml M or E protein; Sino Biological, Beijing, China). Spots were counted with a CTL Immunospot Reader (Cellular Technology Ltd, Cleveland, USA). The result was presented with spot forming cells (SFCs) per million PBMCs or splenocytes.

## Intracellular cytokine staining (ICS) and flow cytometry

Mouse splenocytes ($2\times10^6$ cells/well) or marmoset PBMCs ($1\times10^6$ cells/well) were stimulated for 4h with M or E-derived peptides (ZIKV-M or E protein; Sino Biological, Beijing, China), or medium as negative control. The cells were incubated with Golgi Plug (BD Bioscience, NJ, USA) for 12h at 37˚C, and then reacted for 30min with anti-mouse or anti-human CD3, CD4 and CD8 surface marker antibodies (BD Bioscience, NJ, USA). Cells were washed and fixed with IC Fixation buffer, permeabilized with Permeabilization buffer (eBioscience, CA, USA), and finally stained with anti-mouse or anti-human interferon-γ (IFN-γ), interleukin-2 (IL-2) and tumor necrosis factor α (TNF-α) (BD Bioscience, NJ, USA). The identity of all antibodies was provided in S3 Table. All samples were tested with BD FACSCanton flow cytometer (BD Bioscience, NJ, USA).

## RT-qPCR and RT nested-PCR

ZIKV RNA was purified from marmosets' sera and body fluids (included urine, saliva and feces) using High Pure Viral Nucleic Acid Kit (Roche Diagnostic GmbH, Mannheim, Germany). Viral RNA was quantified by reverse transcription quantitative-PCR (RT-qPCR) assay targeting ZIKV NS5 with the specific primers and probe (S2 Table).

ZIKV in blood or body fluids of marmosets was further identified by RT nested-PCR. The amplicons were sequenced commercially (Beijing Genomics Institute, Beijing, China).

## Detection of viral genome in tissues

Marmosets M47 (immunized with Sad23L-prM-E) and M46 (inoculated with PBS as sham control) were euthanized at day 72 post challenge with ZIKV-Z16006. Animal tissues (brain, testis, lymph nodes, spleen, liver and ovary) were separated to extract ZIKV RNA with Qiagen RNeasy Mini Kit (Qiagen, Hilden, Germany). The viral RNA was quantified by RT-qPCR and viral load was calculated as viral genomes per g of tissue.

## Distribution and expression of Sad23L-prM-E vaccine *in vivo*

Genomic DNA was extracted from homogenized tissue of challenged marmosets by High Pure Viral Nucleic Acid Kit. Adenoviral hexon sequence of Sad23L vector was amplified by the nested PCR.

Viral RNA was isolated from tissues of marmosets with Qiagen RNeasy Mini Kit (Qiagen, Hilden, Germany), and ZIKV E-specific sequence was amplified by RT nested-PCR. The amplicons were sequenced commercially (Beijing Genomics Institute, Beijing, China).

The E protein expression of Sad23L-prM-E vaccine in marmosets was detected by immunofluorescence assay. PBMCs were isolated from marmosets at 4 months post challenge, and stained with anti-ZIKV E antibodies and DAPI. The pictures were taken with Olympus FV1000.

## Adenovirus cross-neutralizing antibody assay

Marmosets' plasmas were tested on HEK-293 cells for neutralization of simian adenovirus 23 or human adenovirus 5 by Sad23L-GFP or Ad5-GFP expressing green fluorescent protein in a plaque reduction assay as previously described [31].

## Histopathological examination

Marmoset tissues were submitted to Guangzhou Huayin Medical Science Company Limited (Guangzhou, China), where tissues were stained with hematoxylin and eosin (H&E) and examined microscopically for histopathological changes.

## Statistical analyses

Data are analyzed with unpaired one-tailed *t* test, one-way AVONA and one-tailed Mann-Whitney U tests. Statistically significant differences are indicated with asterisks (* $P < 0.05$; ** $P < 0.01$ and *** $P < 0.001$). All graphs are generated with GraphPad Prism 7 software.

# Results

## Characterization of Sad23L-prM-E vaccine

In the novel Sad23L vector, the original orf6 within E4 region of SAdV23 was replaced by the corresponding element of Ad5, which massively improved viral propagation. ZIKV vaccine construct (Sad23L-prM-E) contains the Japanese encephalitis virus signal peptide (JE signal) and full-length prM-E genes of ZIKV-Z16006 strain (Fig 1A). The recombinant Sad23L-prM-E virus was rescued from packaging cell HEK-293. A large amount of Sad23L-prM-E vaccines were produced from HEK-293 cell cultures, and further purified and titrated to contain $4.35 \times 10^{11}$ PFU/ml.

The expression of ZIKV E protein was detected in HEK-293, Vero, Huh7.5.1 and marmoset's PBMCs after Sad23L-prM-E virus infection. The bands specific to anti-ZIKV E protein by Western blotting were seen in the vaccine infected cells, but not in the empty Sad23L virus

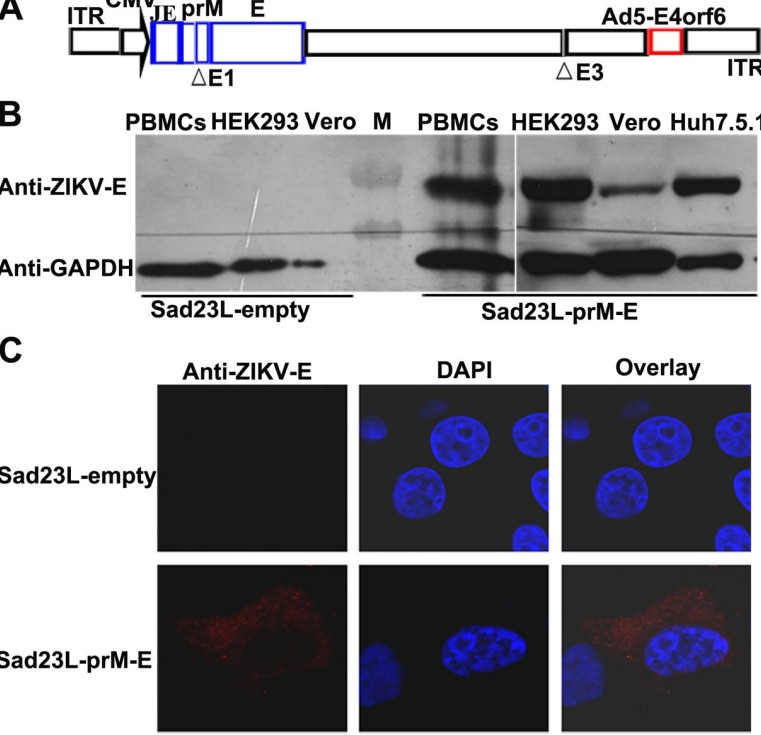

**Fig 1. Characteristics of novel Sad23L-prM-E vaccine.** (A) Genomic construct of Sad23L-prM-E vaccine. Cytomegalovirus promoter (CMV), Japanese encephalitis virus signal peptide (JE signal) sequences and ZIKV prM-E genes were inserted into the deleted E1 region of simian adenovirus type 23 genome (SAdV23), the initial E3 region was deleted and E4orf6 was replaced by the corresponding element of Ad5-E4orf6. ITR indicates inverted terminal repeat sequence. (B) E protein expressions from Sad23L-prM-E virus infected naïve marmoset's PBMCs, HEK-293, Vero and Huh7.1.5 cells were analyzed by Western blot, while Sad23L-empty virus infected cells were used as mock controls. Anti-ZIKV and anti-GAPDH antibodies were used to detect E protein and internal control protein, respectively. M indicates protein marker. (C) E protein expression in Vero cells was detected by immunofluorescence staining.

infected cells (Fig 1B). ZIKV E protein in the cytoplasm of Vero cells infected with Sad23L-prM-E virus was observed by red fluorescence with an immunofluorescence assay, but not in empty vectorial virus (Fig 1C).

## Immunogenicity of Sad23L-prM-E vaccine in mice

To evaluate the immunogenicity of Sad23L-prM-E vaccine, C57BL/6 mice (n = 5/group) were immunized with $5 \times 10^6$, $5 \times 10^7$ or $5 \times 10^8$ PFU Sad23L-prM-E vaccine doses. Control groups (n = 5/group) received $5 \times 10^8$ PFU Sad23L-empty viruses and an equal volume of PBS, respectively. Four weeks post-immunization (S1 Fig), humoral and cellular immune responses were tested. Serum antibody binding to ZIKV E protein (E-Ab) titers were detected in a dose-dependent manner of $10^{2.26}$, $10^{2.73}$ and $10^{3.15}$ from vaccine immunized mice, respectively (Fig 2A), but not from sham control group ($P<0.001$). The neutralizing antibody (NAb) titers to ZIKV equally followed a dose-dependent pattern ($10^{1.78}$, $10^{2.57}$ and $10^{2.95}$) in vaccinated mice (Fig 2B) but not in the sham groups ($P<0.001$). The results indicated that Sad23L-prM-E vaccine induced high titer, dose dependent humoral immune response against ZIKV.

T-cell response of splenocytes from Sad23L-prM-E vaccine immunized mice was detected by ELISpot after stimulation with M or E peptides, respectively (Fig 2C and 2D). Strong specific IFN-γ response to M (66–388 SFCs/million cells) or E peptides (278–786 SFCs/million cells) was measured in a dose-dependent fashion in the vaccine group of mice (Fig 2C and 2D), which was significantly higher than observed in the sham group of mice ($P = 0.0037$, $P<0.001$).

The specific intracellular cytokines of IFN-γ, IL-2 and TNF-α expressing in CD4$^+$ or CD8$^+$ T cells responding to M or E peptides were tested in vaccinated and sham mice (Sad23L-empty and PBS) by ICS. Significantly higher frequency of cytokine-expressing T-cells was observed with $5 \times 10^8$, $5 \times 10^7$ and $5 \times 10^6$ PFU vaccines immunized mice for IL-2$^+$ CD4$^+$ (0.68 ± 0.164%, 0.64 ± 0.195% and 0.26 ± 0.051% vs 0.18 ± 0.089% and 0.17 ± 0.087%, $P = 0.029$, Fig 2E and 2F) and IL-2$^+$ CD8$^+$ cells (0.47 ± 0.103%, 0.36 ± 0.098% and 0.15 ± 0.037% vs 0.08 ± 0.018% and 0.13 ± 0.046%, $P = 0.0025$, Fig 2E and 2G) to M peptides, and IL-2$^+$ CD4$^+$ (0.35 ± 0.037%, 0.26 ± 0.055% and 0.12 ± 0.038% vs 0.04 ± 0.011% and 0.04 ± 0.015%, $P<0.001$, Fig 2H and 2I), IL-2$^+$ CD8$^+$ (0.32 ± 0.082%, 0.18 ± 0.060% and 0.18 ± 0.028% vs 0.06 ± 0.014% and 0.03 ± 0.014%, $P = 0.0032$, Fig 2H and 2J), IFN-γ$^+$ CD4$^+$ (0.53 ± 0.165%, 0.30 ± 0.111% and 0.13 ± 0.017% vs 0.03 ± 0.013% and 0.04 ± 0.016%, $P = 0.0037$, Fig 2K and 2L) and TNF-α$^+$ CD8$^+$ cells (0.19 ± 0.038%, 0.16 ± 0.033% and 0.13 ± 0.021% vs 0.07 ± 0.019% and 0.04 ± 0.014%, $P = 0.0034$, Fig 2M and 2N) to E peptides, respectively. However, the frequency of T-cells was not found statistically different for IFN-γ$^+$ CD4$^+$, IFN-γ$^+$ CD8$^+$, TNF-α$^+$ CD4$^+$ and TNF-α$^+$ CD8$^+$ cells to M peptides; and TNF-α$^+$ CD4$^+$ and IFN-γ$^+$ CD8$^+$ cells to E peptides between vaccinated and control mice (S2 Fig, $P>0.05$).

Taken together, the data indicated that ZIKV prM and E proteins delivered by Sad23L-based vaccine presented strong immunogenicity and induced a robust, specific, humoral and cellular immune response in mice.

## Immune response of marmosets to Sad23L-prM-E vaccine

Baseline values of antibody and T-cell response in pre-vaccination (week 0) were detected individually from the sera and PBMCs of five marmosets. The background mean values of IFN-γ secretion PBMCs to M or E peptides were detected as 13.6 and 7.5 SFCs/million cells by ELISpot, respectively (Fig 3A and 3C). The mean values of frequencies for intracellular IFN-γ$^+$ CD4$^+$ (0.021%) and IFN-γ$^+$ CD8$^+$ (0.041%) cells to M peptides, and intracellular IFN-γ$^+$ CD4$^+$

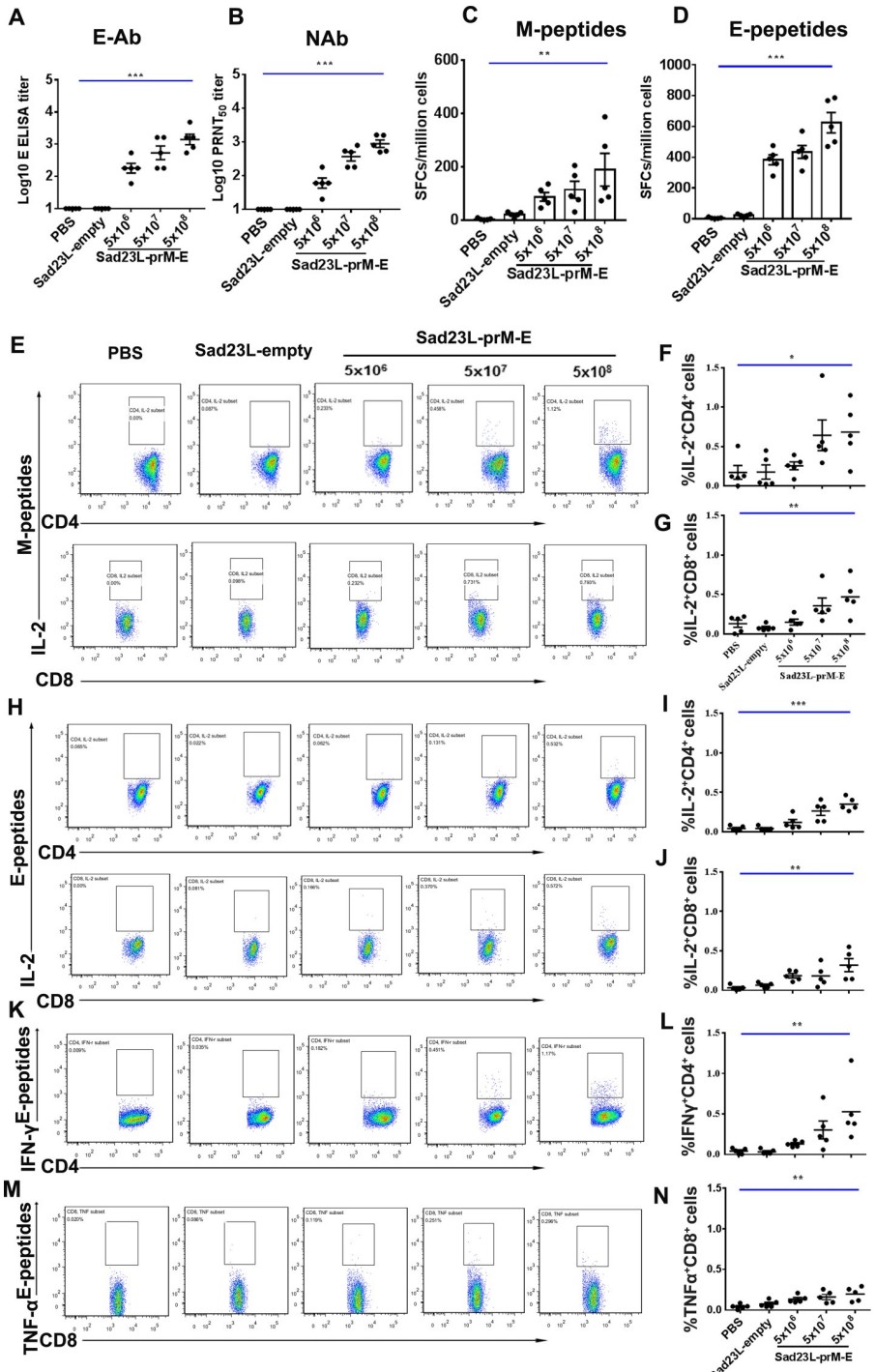

**Fig 2. Humoral and cellular immune dose response of Sad23L-prM-E vaccine to mice.** C57BL/6 mice were immunized by a single dose of $5\times10^6$, $10^7$ or $10^8$ PFU Sad23L-prM-E vaccine, Sad23L-empty or PBS control, respectively. Sera and splenocytes were collected from vaccinated mice for measurement of antibody and T cell responses at 4 weeks post-immunization. (A) E-Ab titer was measured by ELISA. (B) NAb titer was measured by $PRNT_{50}$. (C and D) The number of specific IFN-$\gamma$ spot forming cells (SFCs) per million splenocytes to M or E peptides was measured by ELISpot, respectively. (E-G) Intracellular $IL-2^+$ $CD3^+CD4^+$ and $IL-2^+$ $CD3^+CD8^+$ cells to M peptides were detected by ICS. (H-N) Intracellular $IL-2^+$ $CD3^+CD4^+$, IL-2+ CD3+CD8+, IFN-$\gamma^+$ $CD3^+CD4^+$ and TNF-$\alpha^+$ $CD3^+CD8^+$ T cells to E peptides were detected by ICS, respectively. Data are shown as mean ± SEM (standard errors of means). *P* values are analyzed by one-way ANOVA. Statistically significant differences are shown with asterisks (*, $P<0.05$; **, $P< 0.01$ and ***, $P< 0.001$); ns, no significant difference; $PRNT_{50}$, 50% plaque reduction neutralization test.

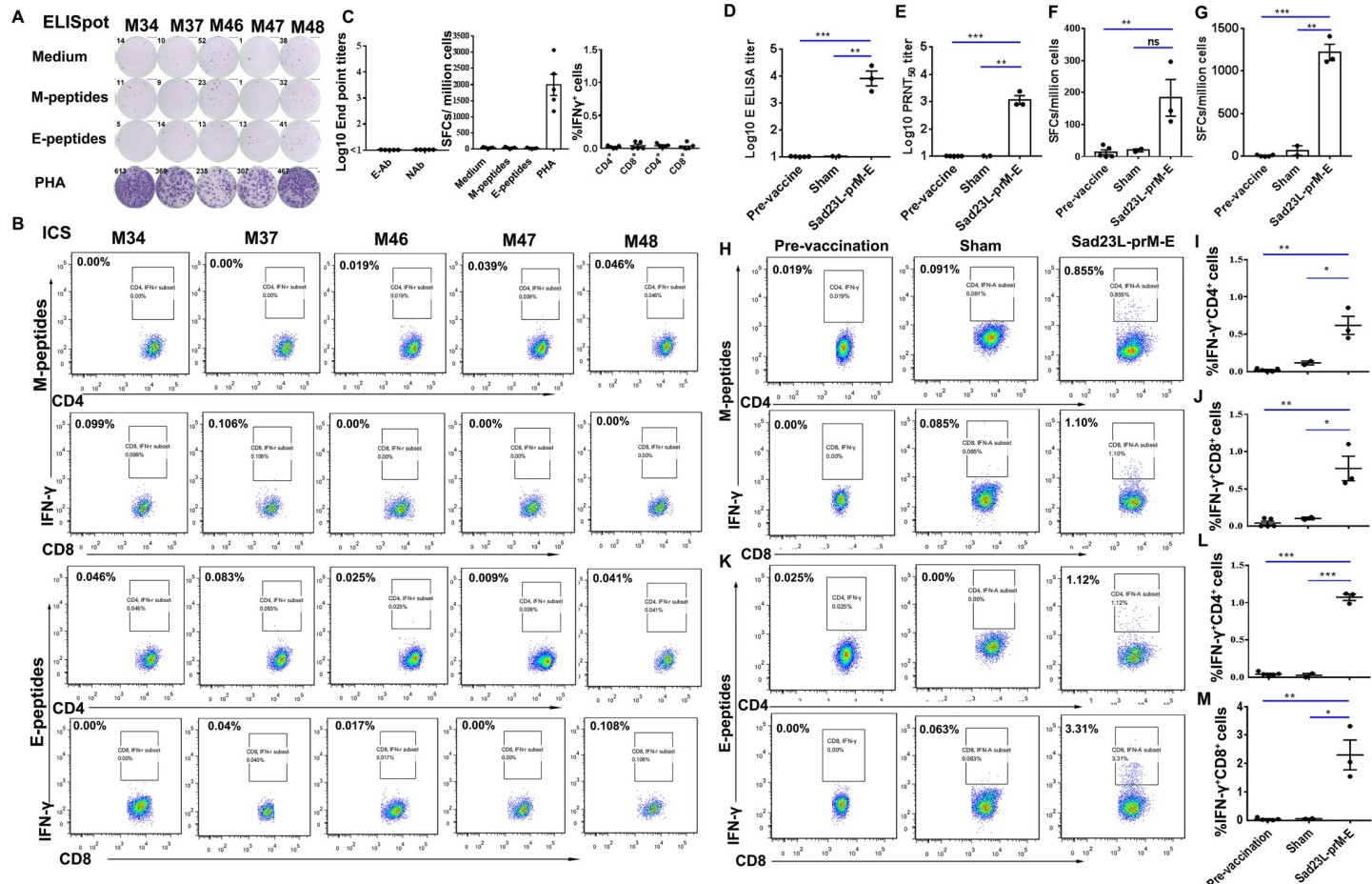

**Fig 3. Humoral and cellular immune response of common marmosets to Sad23L-prM-E vaccine.** Baseline values of antibody and T-cell responses for pre-vaccination were tested individually in the sera and PBMCs of five marmosets (week 0), and a mean value was calculated as a background for each reaction in marmosets. (A) Testing of the background of IFN-γ secreting spot forming cells (SFCs) per million PBMCs to M or E peptides by ELISpot. (B) Testing of the background of intracellular IFN-γ+ CD3+CD4+ or IFN-γ+ CD3+CD8+ cells to M or E peptides by ICS assay. (C) The baseline values of E-binding and neutralizing antibodies and T cell responses in pre-vaccination marmosets. Three marmosets were immunized with a single dose of $3\times10^8$ PFU Sad23L-prM-E vaccine and two marmosets were inoculated with PBS as sham vaccine. From these vaccinated or sham marmosets, sera and PBMCs were isolated to test specific antibody and T-cell responses at 4 weeks post-immunization, and to compare with pre-vaccination values. (D) E-Ab was detected by ELISA. (E) NAb was detected with PRNT_{50}. (F and G) The number of IFN-γ SFCs per million PBMCs to M or E peptides was measured by ELISpot, respectively. (H-J) The percent of intracellular IFN-γ+ CD3+CD4+ or IFN-γ+ CD3+CD8+ cells to M peptides was detected by ICS assay. (K-M) The percentage of intracellular IFN-γ+CD3+CD4+ or IFN-γ+CD3+CD8+ cells to E peptides was detected by ICS assay, respectively. Data are shown as a mean ± SEM (standard errors of means). $P$ values are analyzed with one-tailed $t$ test. Statistically significant differences are shown with asterisks (*, $P<0.05$; **, $P<0.01$ and ***, $P<0.001$). ns, no significant difference.

(0.040%) and IFN-γ+ CD8+ (0.033%) cells to E peptides were measured by ICS assay (Fig 3B and 3C). The baseline values of antibody titers were detected by ELISA and PRNT_{50} as E-Ab <1:10 and NAb <1:10 in the blood of pre-vaccination marmosets, respectively (Fig 3C).

Marmosets were immunized with a single dose of $3\times10^8$ PFU Sad23L-prM-E viruses for the vaccine group (n = 3), and with an equal volume of PBS for the sham group (n = 2). All animals well tolerated the inoculation of a single dose of vaccine and presented no vaccine-related adverse effects. E-Ab reactivity and NAb titer were quantified by ELISA or PRNT_{50} 4 weeks post-inoculation of vaccine or PBS, respectively. The vaccinated marmosets developed E-Ab with a mean endpoint titer of $10^{4.07}$ (Fig 3D), significantly higher than observed in the sham ($P = 0.002$) and pre-vaccination groups ($P<0.001$). The NAb titer reached a mean endpoint of

$10^{3.13}$ for immunized marmosets (Fig 3E), but was not detectable in the sham ($P = 0.0011$) and pre-vaccination marmosets ($P<0.001$).

PBMCs were isolated from whole blood 4 weeks post-vaccination and tested for T cell responses by ELISpot and ICS following stimulation with M or E peptides. The results showed that M peptides induced IFN-γ response with $183.2 \pm 57.42$ SFCs/million cells in the immunized group, a level significantly higher than in pre-vaccination marmosets ($P = 0.0038$), but was not statistically different in the sham group ($P = 0.0573$, Fig 3F). E peptides stimulated strong secretion of IFN-γ ($1,219 \pm 94.4$ SFCs/million cells) in vaccinated marmosets, a level significantly higher than in the sham ($P = 0.0015$) and pre-vaccination marmosets ($P<0.001$, Fig 3G).

The ICS results showed that the percentage of IFN-γ, IL-2 and TNF-α expressing CD4$^+$ or CD8$^+$ T cells was increased in vaccinated marmosets, following stimulation with M peptides (Fig 3H–3J and S3A–S3D Fig), significantly higher frequency of IFN-γ$^+$ CD4$^+$ ($P = 0.0003$, $P = 0.0251$, Fig 3I), IFN-γ$^+$ CD8$^+$ ($P = 0.0005$, $P = 0.0256$, Fig 3J), IL-2$^+$ CD4$^+$ ($P = 0.0013$, $P = 0.0098$, S3A Fig) and TNF-α$^+$ CD4$^+$ cells ($P = 0.0015$, $P = 0.031$, S3C Fig) than that in pre-vaccination and sham group. When stimulated with E peptide, the frequency of IFN-γ producing CD4$^+$ or CD8$^+$ T cells was increased to $1.07 \pm 0.045\%$ and $2.25 \pm 0.679\%$ in the vaccine group (Fig 3K–3M), levels significantly higher than those observed in the sham group ($P<0.001$, $P = 0.0227$) and pre-vaccination group ($P<0.001$, $P = 0.0005$), respectively. The percentage of IL-2$^+$CD4$^+$ T cells to E peptides was increased to $0.17 \pm 0.051\%$ in immunized marmosets (S3E Fig), significantly higher than observed in the sham marmosets ($P = 0.0438$). The percentage of IL-2$^+$ CD8$^+$ T cells to E peptides was increased to $0.12 \pm 0.016\%$ in immunized marmosets (S3F Fig), significantly higher than observed in the sham marmosets ($P = 0.0133$). The percentage of TNF-α$^+$ CD8$^+$ T cells to E peptides was increased to $0.30 \pm 0.083\%$ in immunized marmosets (S3H Fig), significantly higher than observed in the pre-vaccination ($P = 0.0062$) and sham ($P = 0.0439$) marmosets. However, the frequency of IL-2$^+$ CD4$^+$, TNF-α$^+$ CD4$^+$ and IL-2$^+$ CD8$^+$ T cell responses to E peptides in the vaccine group was not statistically different from the sham or pre-vaccination groups ($P>0.05$) (S3E, S3F, S3G Fig).

Overall, the results indicated that Sad23L-prM-E vaccine elicited high levels of E-binding and neutralizing antibodies and robust specific T-cell response against E protein in marmosets 4 weeks after a single dose immunization.

## Protection of Sad23L-prM-E vaccinated marmosets against ZIKV challenge

The vaccinated and non-vaccinated marmoset groups were intramuscularly challenged with a high dose of $1\times10^5$ PFU ZIKV-Z16006 viruses 6 weeks after prime immunization (S1 Fig). Blood and body fluids (saliva, urine and feces) samples were examined daily for viral genome by RT-qPCR, confirmed by RT nested-PCR and sequencing (Fig 4). Data showed that sham marmosets (n = 2) presented with high viral load in plasma, saliva, urine and feces samples (Fig 4A–4D), while the vaccinated marmosets (n = 3) presented transient viral RNA in plasma and body fluids at borderline limit of detection (Fig 4E–4H). The peak viral load in plasma, saliva, urine and feces was compared between sham and vaccinated marmoset groups (Fig 4I–4L). Sham marmosets M46 and M48 presented peak viremia of $10^{5.34}$ or $10^{5.73}$ copies/ml, respectively. In contrast, three vaccinated marmosets M34, M37 and M47 had viral load peaks of $10^{3.67}$, $10^{2.77}$ or $10^{3.33}$ copies/ml, respectively (Fig 4I); 100–1000 folds lower than observed in sham marmosets ($P = 0.0042$). Such low level viremia was only detected by confirmatory test on 1$^{st}$ day post injection for vaccinated M34, 3$^{rd}$ day for vaccinated M37 and M47, suggesting transient viremia following virus inoculation, while high viremia level was confirmed on the 5$^{th}$ day for both non-vaccinated sham marmosets M46 and M48 (Fig 4M). In addition, high

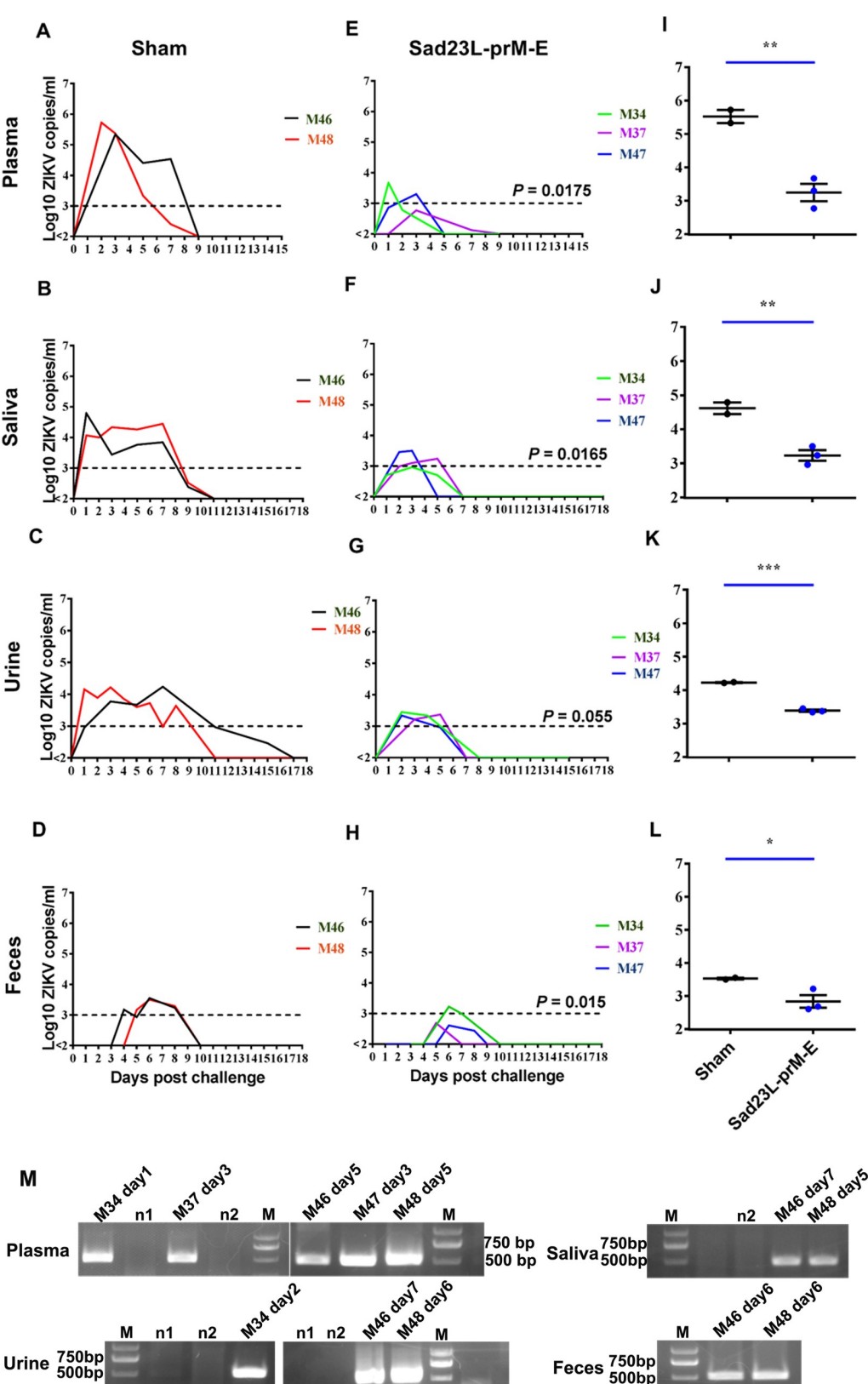

**Fig 4. Protection of Sad23L-prM-E vaccine immunized marmosets against ZIKV infection.** Three marmosets (M34, M37 and M47) were immunized with a single dose of $3\times10^8$ PFU Sad23L-prM-E vaccines, and two marmosets (M46 and

M48) were inoculated with PBS as sham control. All animals were intramuscularly challenged with $10^5$ PFU ZIKV in 6 weeks post vaccination. Sera and body fluids were collected daily after ZIKV challenge for determining viral loads by RT-qPCR and confirming by RT nested-PCR and sequencing. (A-D) Detection of viral load of sham marmosets from plasma, saliva, urine and feces by RT-qPCR. (E-H) Detection of viral load of vaccinated marmosets from plasma, saliva, urine and feces by RT-qPCR. (I-L) Comparison of peak viral loads between sham and vaccinated marmosets in plasma, saliva, urine and feces. (M) Confirmation of ZIKV RNA for predicted amplicons (500 bp) by RT nested-PCR and sequencing was found for plasma, saliva, urine and feces samples from sham marmosets M46 and M48 at 5th to 7th day post challenge, vaccinated M34 at 1st day in plasma and 2nd day in urine, and vaccinated M37 and M47 at 3rd day in plasma, respectively. The dashed line indicated a relatively low level of viral load. $P$ values are analyzed by one-tailed Mann-Whitney U tests or one-tailed $t$ test. Statistically significant differences are shown with asterisks (*, $P<0.05$; **, $P< 0.01$ and ***, $P< 0.001$). ns, no significant difference. n1 and n2 indicate negative control. M indicates DNA molecular markers.

viremia in the two sham marmosets prolonged up to 7 days ($P$ = 0.0175, Fig 4A), suggesting replication of the challenge ZIKV in the non-vaccinated animals. Furthermore, high ZIKV RNA load was detected in saliva ($P$ = 0.0051, Fig 4J), in urine ($P$ = 0.0001, Fig 4K) and in feces ($P$ = 0.0346, Fig 4L), with viral identification at 1st to 9th days in saliva ($P$ = 0.0165, Fig 4B), 4th to 8th days in feces ($P$ = 0.015, Fig 4D) and 1st to 11th days in urine ($P$ = 0.055, Fig 4C) samples of sham marmosets. Significantly lower viral load and shorter viral detection was found in vaccinated marmosets (Fig 4E–4L). RT nested-PCR amplicons of 500bp DNA bands were confirmed by sequencing for ZIKV RNA only on post-challenge 2nd day urine sample of vaccinated M34, but on the 5th to 7th day in saliva, urine and feces samples of sham marmosets M46 and M48 (Fig 4M). This data evidenced that challenged ZIKV replicated and persisted in non-vaccinated marmosets but not in vaccinated marmosets.

## Follow-up examination for clinical manifestation and pathology of ZIKV challenged marmosets

Sham marmoset M48 was found to lose body weight at 9th day post challenge. The lowest weight of 73% of initial was found at 22nd day and went back to 89% at 44th day post viral challenge. Other marmosets had no obvious change of weight (S4 Fig).

Sham M46 and vaccinated M47 marmosets were euthanized at 72nd day post-challenge (S1 Fig). Several types of tissues including brain, testis, lymph node, spleen, liver and ovary did not contain ZIKV RNA by RT-qPCR (S4 Table). No significantly pathological lesions were observed in any of these tissues from both animals (S5 Fig). Sham M48 and vaccinated M37 died at 290th and 349th day post-challenge, respectively, and their testis were isolated and detected negative for ZIKV RNA. Sham M48 displayed a smaller size and lighter weight of testis compared to both vaccinated M47 and M37 (S6A Fig), but his histopathological examination of testis did not show overt tissue damage compared to both vaccinated M47 and M37 (S6B Fig).

Body fluids (serum, saliva, urine and feces) of vaccinated M34 and M37 and sham M48 marmosets were continuously examined for four months after ZIKV challenge. None of the collected samples were positive for ZIKV by RT-qPCR beyond 17th day post ZIKV challenge in all marmosets (Fig 4C).

## Secondary immune response and its duration in vaccinated marmosets after ZIKV challenge

The prime vaccinated marmosets were tested for secondary humoral and cellular immune response after ZIKV challenge in comparison with sham marmosets (Fig 5). Titer of NAb reactivity in vaccinated marmosets rapidly increased to levels above $10^{3.66}$ at 1st-2nd week post-challenge and was sustained at least up to 13th week (Fig 5A), while NAb response in sham

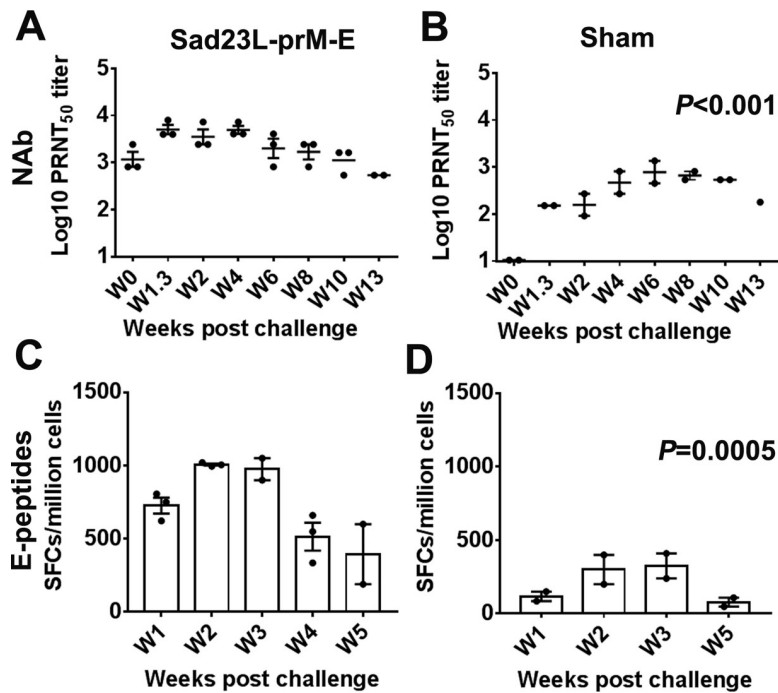

**Fig 5. Memory immune response in vaccinated marmosets against ZIKV infection over time.** Vaccinated marmosets were tested for secondary neutralizing antibody and T-cell immune response after exposure to ZIKV challenge in comparison with sham marmosets. Titration of serum neutralizing antibody from vaccinated marmosets by $PRNT_{50}$; (A) in vaccinated animals, (B) in control unvaccinated animals. Measurement of IFN-γ secreting T cell response of PBMCs to E peptides by ELISpot; (C) in vaccinated marmosets, (D) in non-vaccinated sham marmosets. The number of IFN-γ spot forming cells (SFCs) per million PBMCs is calculated in ELISpot. Data are shown as a mean ± SEM (standard errors of means). Statistically significant differences of neutralizing antibody (NAb) or T cell response (E-peptides) between vaccination and sham groups are compared by one-tailed Mann-Whitney U tests, and the P values are presented in Fig 5B and 5D, respectively.

marmosets slowly reached a peak of $10^{2.91}$ at 6th week and then declined to $10^{2.26}$ at 13th week after virus challenge (Fig 5B), the difference was significant between the two groups ($P<0.001$).

Similar pattern for specific T-cell response to E peptides was observed between vaccinated and non-vaccinated marmosets after ZIKV challenge (Fig 5). The vaccinated marmosets had quick and strong secondary IFN-γ secreting T-cell response to ZIKV (up to 1,006 SFCs/million cells; Fig 5C), while the sham marmosets had significantly lower T-cell response (up to 325 SFCs/million cells; Fig 5D, $P = 0.0005$). Data suggested that a single dose of vaccine in marmosets could elicit strong memory protective immune response when exposure to ZIKV.

## Delivery and expression of Sad23L-prM-E vaccine in marmoset's tissues

Pre-existence of neutralizing antibody to Sad23L or Ad5 vectorial adenovirus was tested for all involved marmosets before inoculation of vaccine. Animals showed no reactivity to Sad23L, but one (M34) had 1:40 NAb titer to Ad5 (S1 Table). After PBS injection and ZIKV challenge, two sham marmosets M46 and M48 had no detectable NAb to Sad23L vector (Fig 6A). As expected, three vaccinated marmosets M34, M37 and M47 produced high levels of NAb specific to Sad23L vector after a single dose of Sad23L-prM-E vaccine inoculation (Fig 6A), which confirmed this vector worked well even though M34 had higher NAb reactivity to Ad5.

Spleen, lung, kidney, liver and muscle tissues (at intramuscular injection site and para-tissues) from sham M46 and vaccinated M47 at 72nd day after ZIKV challenging were examined

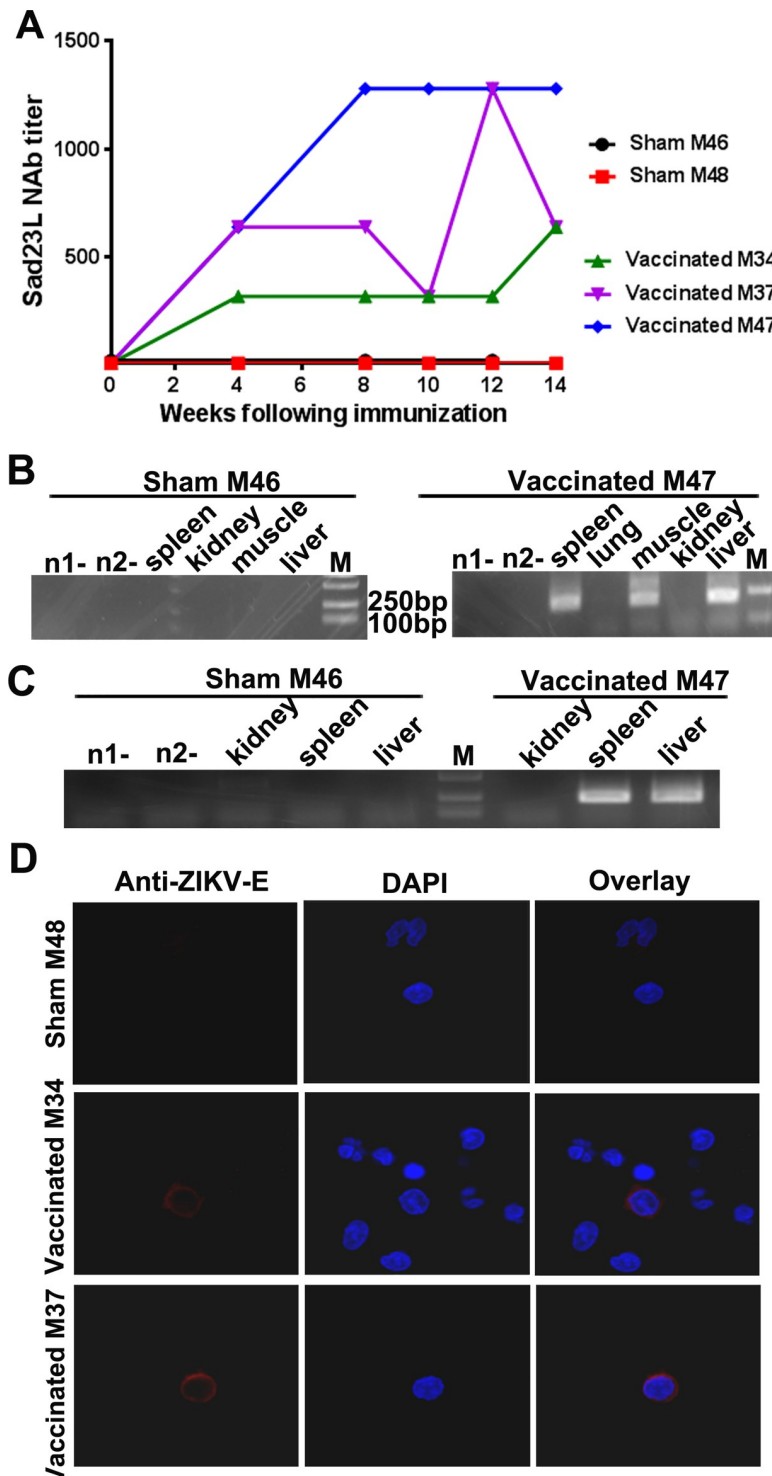

**Fig 6. Biodistribution and expression of Sad23L-prM-E vaccine in marmosets.** (A) Neutralizing antibody titers to Sad23L vectorial virus after prime immunization with Sad23L-prM-E vaccine. (B) Nested-PCR amplifying Sad23L-hexon gene (200bp) in tissues (spleen, lung, muscle, kidney and liver) at 16 weeks after prime immunization with Sad23L-prM-E vaccine. (C) RT nested-PCR amplifying ZIKV E mRNA transcripts in tissues (spleen, muscle, kidney and liver) of marmosets at 10 weeks (72 days) post challenge. (D) Immunofluorescence staining to detect the expression of E protein in PBMCs of immunized marmosets. PBMCs were isolated from marmosets in 4 months post challenge, stained by ZIKV E antibody and DAPI. Red immunofluorescence of ZIKV E protein was detected in vaccinated marmosets M34 and M37 but not in non-vaccinated sham marmosets M48.

for assessment of vaccine delivery by amplifying Sad23L vector sequence with primers specific for the adenoviral hexon. Spleen, muscle and liver of vaccinated animal M47 presented the predicted size of Sad23L vectorial DNA bands, while tissues of sham M46 were all negative (Fig 6B).

To determine the expression of Sad23L-prM-E vaccine in marmosets, viral RNA was extracted from tissues (kidney, spleen and liver) and amplified for the RNA transcripts of E gene by RT nested-PCR. Predicted 250bp DNA fragments of E reverse transcripts were observed in tissues of spleen and liver from the vaccinated M47, but not in tissues from sham M46 (Fig 6C). In addition, the immunofluorescence staining focused the E antigen expression of Sad23L-prM-E vaccine in PBMCs in vaccinated marmosets M34 and M37 four months post challenge, but was not seen in sham marmoset M48 (Fig 6D). These results indicated that Sad23L-prM-E vaccine could transduce splenocytes, hepatocytes as well as PBMCs in marmosets, in whom the vaccine expressed and sustained ZIKV antigens, and elicited persistent stimulation of host immunity.

## Discussion

In this study, we generated a ZIKV vaccine on basis of our newly improved simian adenovirus type 23 vector (Sad23L), and examined the efficacy of vaccine depending on the following principals:

Firstly, the immunogenicity of prM-E proteins was previously shown to elicit protective immunity against ZIKV infection in mice and rhesus monkeys [18,21,32,33]. The E protein in particular is the major protein involved in ZIKV receptor binding and fusion [34], which can be blocked by neutralizing antibody and cellular response [35,36].

Secondly, simian adenovirus type 23 (SAdV23 or AdC6) was a low-seroprevalence strain in humans. Recombinant vector was therefore constructed for use in vaccine development [37–40], but the low infectious titer was found in these studies would impact vaccine production *in vitro* and transduction *in vivo*. Within the improved Sad23L vector, the original orf6 in E4 region was replaced by the corresponding element of Ad5, which massively enhanced the infectious titer up to 870 PFU per a single packaging cell [27]. The immunity of Sad23L-prM-E vaccines was not affected by the pre-existing Ad5 neutralizing antibody (titer 1:40) in marmoset M34, which confirmed that Sad23L could escape from Ad5 neutralizing antibody influence.

Thirdly, non-human primates (NHP) are the natural reservoirs of ZIKV and are considered the most suitable pre-clinical models for Zika disease [41,42]. However, ethical restrictions and maintenance cost for NHP are high, which limits their use for development of vaccines [43]. Common marmosets are new world small primate utilized as central animal model for preclinical and translational medicine of human biology and disease [23,24]. In addition, we have accumulated considerable experience with this small model when developing our HCV/GBV-B chimera infected marmosets [44,45].

In this study, a single dose of $5\times10^6$, $5\times10^7$ or $5\times10^8$ PFU Sad23L-prM-E vaccines elicited high levels of NAb and IFN-γ secreting T-cell response to prM-E antigens in a dose dependent manner in mice, suggesting that the vaccine had strong specific immunogenicity, comparable with previous studies [18,46]. Furthermore, a single dose of $3\times10^8$ PFU Sad23L-prM-E vaccine was chosen to vaccinate marmosets primely, inducing long-lasting memory immunity, and the strong secondary NAb (titer $>10^{3.66}$) and T-cell immune response (726 SFCs/$10^6$ PBMCs) following exposure to ZIKV challenge would rapidly eradicate ZIKV in vaccinated marmosets, which was advanced over other adenovirus vectorial ZIKV vaccines such as RhAd52 [21], Ad26 [47], Ad5 [48] and AdC7 [22].

A low level of transient viremia or excreted viruses were detected in body fluids (saliva, urine and feces) at a borderline level in vaccinated marmosets upon challenge with $10^5$ PFU ZIKV, which corresponded to more than 100-fold reduction of viral RNA load compared with results obtained with non-vaccinated control marmosets ($P<0.05$). The virus in non-vaccinated sham marmosets replicated up to a level of $10^{5.73}$ copies/ml and persisted much longer, but decreased on the 9th day (week1.3) post challenge when NAb increased. The ZIKV in blood or body fluids was considered transient excretion in vaccinated animals following intramuscular challenge, which did not replicate but was rapidly cleared in immunized marmosets without pathological lesion. This similar pattern was observed in immunized rhesus macaques receiving a single dose of DNA vaccine (1mg) followed by a $10^3$ PFU ZIKV challenge [17], or in vaccinated rhesus macaques receiving a single dose of DNA or purified inactivated virus (PIV) vaccine followed by a $10^3$ PFU ZIKV challenge [49], respectively.

The relatively low cost of maintaining and easily handling for marmosets allowed us to conduct a long period of follow-up investigation on the vaccinated marmosets after ZIKV challenge. Urine was considered as the first choice followed by saliva for monitoring ZIKV excretion in marmosets as it had a higher viral load and longer persistence with easy sampling. The vector genome and ZIKV E gene transcripts of Sad23L-prM-E vaccine were sustained in spleen and liver 10 weeks, and E protein expression was observed in PBMCs 4 months post challenge, while the specific NAb and IFN-γ secreting T-cell responses were maintained at high levels at the endpoint of this study. The results indicated that the prime immunization with Sad23L-prM-E vaccine was effectively delivered and that sufficient protective immunity was acquired by the host against ZIKV infection.

In conclusion, a prime immunization with Sad23L-prM-E was able to induce strong protective neutralizing antibody and robust T-cell immune response against ZIKV infection in common marmosets. The efficacy of prime vaccination could eradicate $10^3$–$10^4$ PFU ZIKV challenge, and further rapidly sterilize the extra viruses from vaccinated marmosets when exposed to a higher amount of viruses up to $10^5$ PFU, which suggested that the novel Sad23L-prM-E vaccine was a promising vaccine candidate for clinical trial and potentially preventing ZIKV infection in human.

## Supporting information

**S1 Fig. Schedules for vaccination, detection and challenge.** Mice and common marmosets were immunized with Sad23L-prM-E at week 0; evaluation for immunogenicity at 4 weeks post vaccination. Marmosets were challenged with ZIKV at 6 weeks post vaccination; Sham M46 and vaccinated M47 were euthanized at 72nd day post challenge; Sham M48, vaccinated M34 and M37 were persistently monitored.
(TIF)

**S2 Fig. The rate of intracellular cytokine detection in splenocytes from mice immunized with a different single dose of Sad23L-prM-E vaccine 4 weeks post immunization, which was not statistically different from non-immunized PBS or Sad23L-empty control mice.** (A-D) The rate of intracellular cytokine+ CD4+ or CD8+ cells of splenocytes to M peptides. (E and F) The rate of intracellular cytokine+ CD4+ or CD8+ cells of splenocytes to E peptides. Data is shown as means ± SEM (standard errors of means). $P$ values are analyzed by one-way ANOVA. Statistically significant differences are showed with asterisks (*, $P<0.05$; **, $P< 0.01$ and ***, $P< 0.001$). ns, $P>0.05$ and no significant difference.
(TIF)

**S3 Fig. The rate of intracellular cytokine detection in PBMCs from marmosets immunized with a single dose of Sad23L-prM-E vaccine in 4 weeks post vaccination.** (A-D) The rate of intracellular cytokine$^+$ CD3$^+$CD4$^+$ or CD3$^+$CD8$^+$ cells of PBMCs to M peptides. (E-H) The rate of intracellular cytokine$^+$ CD3$^+$CD4$^+$ or CD3$^+$ CD8$^+$ cells of PBMCs to E peptides. Data is shown as means ± SEM (standard errors of means). $P$ values are analyzed with one-tailed $t$ test. Statistically significant differences are showed with asterisks (*, $P<0.05$; **, $P< 0.01$ and ***, $P< 0.001$). ns, $P>0.05$ and no significant difference.
(TIF)

**S4 Fig. Monitoring of marmosets' weights post ZIKV challenge.** Weight loss was only observed in non-vaccinated M48 and began at 9th day post challenge. The lowest weight was 73% of initial weight at 22nd day, and then went back to 89% of initial weight at 44th day post ZIKV challenge.
(TIF)

**S5 Fig. Histopathological observation of four types of tissues from sham and vaccinated marmosets post ZIKV challenge.** Marmoset tissues were isolated and immediately fixed in 10% buffered formalin solution. The tissues were stained with hematoxylin and eosin (H&E), and examined microscopically for histopathological changes at a magnification of 100 ×.
(TIF)

**S6 Fig. Pathological and histopathological examinations of testis from vaccinated and sham marmosets post ZIKV challenge.** Vaccinated M47 was euthanized at 72nd day post-challenge. Sham M48 and vaccinated M37 died at 290th and 349th day post-challenge, respectively. Marmoset testis were isolated and stained with hematoxylin and eosin (H&E) and examined microscopically for histopathological changes. (A) Testis from marmosets shown in size and weight. (B) Images of testes histopathology.
(TIF)

**S1 Table. Common marmoset immunization regimen.**
(DOCX)

**S2 Table. Primers of RT qPCR or RT nested-PCR.**
(DOCX)

**S3 Table. Key reagents and resources.**
(DOCX)

**S4 Table. Detection of ZIKV RNA loads in all types of tissues of marmosets post challenge.**
(DOCX)

## Acknowledgments

The authors thank Mr Jinhui Lu, Qi Wang, Jianhai Yu and Zhiran Qin and Ms Jiawen Wang for their assistance in sample collection and virus culture. The authors thank professor Dongming Zhou from The Institut Pasteur in Shanghai for his supervising of simian adenovirus vector construction.

## Author Contributions

**Conceptualization:** Shengxue Luo, Wei Zhao, Tingting Li, Chengyao Li.

**Formal analysis:** Yongshui Fu, Jean-Pierre Allain, Tingting Li, Chengyao Li.

**Funding acquisition:** Tingting Li, Chengyao Li.

**Investigation:** Shengxue Luo, Xiaorui Ma, Panli Zhang, Bochao Liu, Ling Zhang, Wenjing Wang, Yuanzhan Wang.

**Resources:** Wei Zhao, Tingting Li, Chengyao Li.

**Writing – original draft:** Shengxue Luo, Tingting Li, Chengyao Li.

**Writing – review & editing:** Shengxue Luo, Jean-Pierre Allain, Tingting Li, Chengyao Li.

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
