## [Decision Letter · Decision Letter 0]

12 Oct 2019

Dear Prof. Li:

Thank you very much for submitting your manuscript "A high infectious simian adenovirus type 23 vector based vaccine efficiently protects common marmosets against Zika virus infection" (#PNTD-D-19-01220) for review by PLOS Neglected Tropical Diseases. Your manuscript was fully evaluated at the editorial level and by independent peer reviewers. The reviewers appreciated the attention to an important problem, but raised some substantial concerns about the manuscript as it currently stands. These issues must be addressed before we would be willing to consider a revised version of your study. We cannot, of course, promise publication at that time.

We therefore ask you to modify the manuscript according to the review recommendations before we can consider your manuscript for acceptance. Your revisions should address the specific points made by each reviewer. 

When you are ready to resubmit, please be prepared to upload the following:

(1) A letter containing a detailed list of your responses to the review comments and a description of the changes you have made in the manuscript.

(2) Two versions of the manuscript: one with either highlights or tracked changes denoting where the text has been changed (uploaded as a "Revised Article with Changes Highlighted" file); the other a clean version (uploaded as the article file).

(3) If available, a striking still image (a new image if one is available or an existing one from within your manuscript). If your manuscript is accepted for publication, this image may be featured on our website. Images should ideally be high resolution, eye-catching, single panel images; where one is available, please use 'add file' at the time of resubmission and select 'striking image' as the file type. 

Please provide a short caption, including credits, uploaded as a separate "Other" file. If your image is from someone other than yourself, please ensure that the artist has read and agreed to the terms and conditions of the Creative Commons Attribution License at http://journals.plos.org/plosntds/s/content-license (NOTE: we cannot publish copyrighted images). 

(4) If applicable, we encourage you to add a list of accession numbers/ID numbers for genes and proteins mentioned in the text (these should be listed as a paragraph at the end of the manuscript). You can supply accession numbers for any database, so long as the database is publicly accessible and stable. Examples include LocusLink and SwissProt.

(5) To enhance the reproducibility of your results, we recommend that you deposit your laboratory protocols in protocols.io, where a protocol can be assigned its own identifier (DOI) such that it can be cited independently in the future. For instructions see http://journals.plos.org/plosntds/s/submission-guidelines#loc-methods

While revising your submission, please upload your figure files to the Preflight Analysis and Conversion Engine (PACE) digital diagnostic tool, https://pacev2.apexcovantage.com/ PACE helps ensure that figures meet PLOS requirements. To use PACE, you must first register as a user. Then, login and navigate to the UPLOAD tab, where you will find detailed instructions on how to use the tool. If you encounter any issues or have any questions when using PACE, please email us at figures@plos.org.

We hope to receive your revised manuscript by Dec 11 2019 11:59PM. If you anticipate any delay in its return, we ask that you let us know the expected resubmission date by replying to this email.

To submit a revision, go to https://www.editorialmanager.com/pntd/ and log in as an Author. You will see a menu item call Submission Needing Revision. You will find your submission record there. 

Sincerely,

David W.C. Beasley

Associate Editor

Sunit Singh

Deputy Editor

Reviewer's Responses to Questions

**Key Review Criteria Required for Acceptance?**

**Methods**

-Are the objectives of the study clearly articulated with a clear testable hypothesis stated?

-Is the study design appropriate to address the stated objectives?

-Is the population clearly described and appropriate for the hypothesis being tested?

-Is the sample size sufficient to ensure adequate power to address the hypothesis being tested?

-Were correct statistical analysis used to support conclusions?

-Are there concerns about ethical or regulatory requirements being met?

Reviewer #1: The methods are appropriate for the objective and hypothesis. I have minor comments regarding the statistics.

Reviewer #2: See 'General Comments' below.

**Results**

-Does the analysis presented match the analysis plan?

-Are the results clearly and completely presented?

-Are the figures (Tables, Images) of sufficient quality for clarity?

Reviewer #1: The analysis is appropriate for the hypothesis. The results, figures, and tables are clear.

Reviewer #2: See 'General Comments' below.

**Conclusions**

-Are the conclusions supported by the data presented?

-Are the limitations of analysis clearly described?

-Do the authors discuss how these data can be helpful to advance our understanding of the topic under study?

-Is public health relevance addressed?

Reviewer #1: The analysis and data support the conclusion.

Reviewer #2: See 'General Comments' below.

**Editorial and Data Presentation Modifications?**

Reviewer #1: The manuscript is easy to follow and the data are explained in the figures and tables within the main text and supplement. I have a few minor comments:

1. Comment on choice of animal sex for the study: all mice were female but marmosets were male (with the exception of one control).

2. Please provide additional information regarding the JE signal peptide used. Was the sequence from an attenuated JE virus?

3. Why was strain Z16006 selected for the study? What is its source and history?

4. Why was the dose of 3x10^8 pfu selected for vaccination? 

5. The adult marmoset model is for subclinical disease; therefore, it limits the ability to test whether or not the vaccine reduces disease. 

6. Comment on the weight loss of only one of two sham marmosets. Is the weight loss considered significant in these animals? Were there any other major differences in the response to the challenge between the two sham marmosets?

7. The methods state that viral genome in marmoset tissues was assessed in brain, testis, lymph nodes, spleen, liver, and ovary, but data from all of these tissues are not shown. Please revise accordingly. 

8. No H&E images are included in the manuscript. Was any histopathology expected in these animals? 

9. Please edit the manuscript for typographical and minor language errors. Examples:

“images” not “photos” 

Line 453: “Decreased” not “disappeared” 

“are shown” not “are showed” 

The “10” base is missing after “log” in the y-axes

10. The statistical methods used for each assay should be clarified. Specifically, two-tailed tests are used throughout the study, but are these appropriate for all of the comparisons? One-tailed tests carry more power. Also, please clarify which specific values are included in the comparisons. 

The Figure 4 legend states that the values (points) in Figure 4 I-L represent the peak titers in the preceding panels. What are the p-values in Figure 4 E-K comparing? 

Figure 5- what is being compared to arrive at the p-values in the graphs? 

Fig 4 – It seems that 2log10 is the limit of detection, so what does the dashed line at 3log10 represent?

Reviewer #2: See 'General Comments' below.

**Summary and General Comments**

Reviewer #1: In the manuscript the authors use a simian adenovirus vector vaccine, Sad23-prME, to immunize mice and marmosets against the prME of Zika virus in order to overcome the lower yield issues of another adenovirus vectored ZIKV vaccine using RhAd52. Vaccination of marmosets led to immunity against ZIKV and reduced viral loads (viral RNA) detected in saliva and blood samples following challenge with an Asian ZIKV strain. 

Weaknesses: 

Vaccination did not provide sterilizing immunity, and the narrow spectrum of disease in the marmoset model rather limits assessment of the vaccine’s ability to protect from symptomatic disease that could occur following ZIKV infection. Further, only male marmosets were vaccinated, and it remains to be seen whether female marmosets generate similar response that can protect in a pregnancy model. 

Strengths: 

The authors generated a new vaccine for ZIKV. Immunity in mice was shown prior to moving to NHP experiments. The authors performed a comprehensive set of experiments, many of them with serial sampling, and a wide-range of assays to show that Sad23-prME vaccine elicits antibody and cellular immunity against ZIKV. 

Reviewer #2: Luo et al present a study to highlight the efficacy of a recombinant simian adenovirus serotype 23 vector encoding PrM and E against Zika virus and the utility of marmosets as a challenge model to assess Zika virus vaccine efficacy. Whilst the study presents useful results to advance the knowledge base for Zika virus vaccine development, there are several caveats about the study that need to be addressed:

1) The marmoset vaccine efficacy study does not appear to be sufficiently powered to generate statistically significant results. I am not sure how the authors show p values (less than 0.05) when the sham group only has 2 marmosets for comparison purposes against the vaccinated group. 

2) There is no pre-bleed data presented for the marmoset study to establish a reasonable baseline for T cell and antibody responses for each marmoset. Given that these ar,e I presume, outbred marmosets unlike inbred mice, this data is very relevant for comparison (pre-bleed vs post-vaccination bleeds) unless the authors can justify otherwise. 

3) The frequency data for T cell responses is not convincing because proportion analysis is susceptible to skewing effects. The authors should show representative flow cytometry plots for the ICS analysis in figure 2 and show the absolute cell numbers to show the total number of cytokine producing cells for all the ICS data in the manuscript. Statistical significance comparison in figure 2 should compare all groups analysed and not just the highest dose.

4) Why were the marmosets challenged using the intramuscular route? The data regarding which route is most robust in establishing Zika virus infection of marmosets is lacking. Studies using rhesus macaques show infectivity of Zika virus using the subcutaneous route and is the route used to evaluate vaccine efficacy in macaques even in the relevant papers the authors have cited. 

5) Why were marmosets vaccinated using 3 x 108 PFU of the vaccine when in mice a dose of 5 x 108 PFU elicited the highest immune responses? Based on the mice data, the authors should have used a higher dose which may have resulted in better protective and immune outcomes especially given that some of the T cell responses (i.e. IL-2 and TNF production) in vaccinated marmosets were comparable to the sham control. 

6) The methods section needs to be more descriptive. For example, how many cells were stimulated using peptides? How were the cells cultured and serum added in the PRNT assay? What is composition of ‘complete’ DMEM? If the virus was propagated using HEK 293 cells, how many passages can the recombinant virus be cultured for? The clones of the monoclonal antibodies including those used for ICS should be mentioned. 

7) There are some typos and the second last sentence of the Abstract (starting at line 40) should be re-written because it is confusing. In line 269, replace the word ‘induction’ with stimulation. The abbreviations for IL-2, TNF and IFN need to be spelled out before they are abbreviated initially.

PLOS authors have the option to publish the peer review history of their article (what does this mean?). If published, this will include your full peer review and any attached files.

Reviewer #1: No

Reviewer #2: No

---

## [Decision Letter · Decision Letter 1]

17 Dec 2019

Dear Prof. Li:

Thank you very much for submitting your manuscript "A high infectious simian adenovirus type 23 vector based vaccine efficiently protects common marmosets against Zika virus infection" (PNTD-D-19-01220R1) for review by PLOS Neglected Tropical Diseases. Your manuscript was fully evaluated at the editorial level and by independent peer reviewers. Although the reviewers agreed that the manuscript has been significantly improved, some aspects of their original critiques were not adequately addressed. In particular, it is recommended, and the editors agree, that use of one-tailed rather than two-tailed tests for comparisons of some data be included (see reviewer comments below) given that specific outcomes - improved survival and increased neutralizing antibody titers - should be expected following effective immunization. Additional clarification of the rationale for selecting the dose in marmosets based on testing in mice is also requested. The suggestion in your responses to the original critiques that C57BL/6 mice weighed 50g does not seem correct; weights of approximately 20g for those mice are more typical and would provide a somewhat different basis for proportional dose selection in marmosets. 

We therefore ask you to modify the manuscript according to the review recommendations before we can consider your manuscript for acceptance. Your revisions should address the specific points made by each reviewer.

(1) A letter containing a detailed list of your responses to the review comments and a description of the changes you have made in the manuscript.

(2) Two versions of the manuscript: one with either highlights or tracked changes denoting where the text has been changed (uploaded as a "Revised Article with Changes Highlighted" file ); the other a clean version (uploaded as the article file).

(3) If available, a striking still image (a new image if one is available or an existing one from within your manuscript). If your manuscript is accepted for publication, this image may be featured on our website. Images should ideally be high resolution, eye-catching, single panel images; where one is available, please use 'add file' at the time of resubmission and select 'striking image' as the file type. 

Please provide a short caption, including credits, uploaded as a separate "Other" file. If your image is from someone other than yourself, please ensure that the artist has read and agreed to the terms and conditions of the Creative Commons Attribution License at http://journals.plos.org/plosntds/s/content-license (NOTE: we cannot publish copyrighted images). 

(4) Appropriate Figure Files 

Please remove all name and figure # text from your figure files upon submitting your revision. Please also take this time to check that your figures are of high resolution, which will improve both the editorial review process and help expedite your manuscript's publication should it be accepted. Please note that figures must have been originally created at 300dpi or higher. Do not manually increase the resolution of your files. For instructions on how to properly obtain high quality images, please review our Figure Guidelines, with examples at: http://journals.plos.org/plosntds/s/figures

While revising your submission, please upload your figure files to the Preflight Analysis and Conversion Engine (PACE) digital diagnostic tool, https://pacev2.apexcovantage.com/ PACE helps ensure that figures meet PLOS requirements. To use PACE, you must first register as a user. Then, login and navigate to the UPLOAD tab, where you will find detailed instructions on how to use the tool. If you encounter any issues or have any questions when using PACE, please email us at figures@plos.org.

We hope to receive your revised manuscript by Feb 15 2020 11:59PM. If you anticipate any delay in its return, we ask that you let us know the expected resubmission date by replying to this email.

To submit your revised files, please log in to https://www.editorialmanager.com/pntd/

Sincerely,

David W.C. Beasley

Associate Editor

Sunit Singh

Deputy Editor

Reviewer's Responses to Questions

**Key Review Criteria Required for Acceptance?**

**Methods**

-Are the objectives of the study clearly articulated with a clear testable hypothesis stated?

-Is the study design appropriate to address the stated objectives?

-Is the population clearly described and appropriate for the hypothesis being tested?

-Is the sample size sufficient to ensure adequate power to address the hypothesis being tested?

-Were correct statistical analysis used to support conclusions?

-Are there concerns about ethical or regulatory requirements being met?

Reviewer #1: The authors addressed the comments in the initial review. However, they should perform one-tailed tests and provide the actual p-values, even inf they're greater than 0.05. This is the accepted statistical method to use. A vaccine should increase antibody titers and reduce viremia. Assuming that the data distribution could be two-tailed would imply that titers can be less than zero, which they cannot.

Reviewer #2: (No Response)

**Results**

-Does the analysis presented match the analysis plan?

-Are the results clearly and completely presented?

-Are the figures (Tables, Images) of sufficient quality for clarity?

Reviewer #1: Results are good. Additional figures were included in the supplement.

Reviewer #2: (No Response)

**Conclusions**

-Are the conclusions supported by the data presented?

-Are the limitations of analysis clearly described?

-Do the authors discuss how these data can be helpful to advance our understanding of the topic under study?

-Is public health relevance addressed?

Reviewer #1: Conclusions are good.

Reviewer #2: (No Response)

**Editorial and Data Presentation Modifications?**

Reviewer #1: Some figure panels are very small and could be enlarged.

Reviewer #2: (No Response)

**Summary and General Comments**

Reviewer #1: Overall, the vaccinated animals generated a response to the improved vaccine. Again, it is understandable that small numbers of animals were used for the study, and it can be more difficult to achieve statistical significance with small groups. This is one of the disadvantages of using NHPs, and perhaps the scientific community overly emphasizes statistical significance.

Reviewer #2: (No Response)

PLOS authors have the option to publish the peer review history of their article (what does this mean?). If published, this will include your full peer review and any attached files.

Reviewer #1: No

Reviewer #2: No

---

## [Editor Report · Decision Letter 2]

3 Jan 2020

Dear Prof. Li,

We are pleased to inform you that your manuscript, "A high infectious simian adenovirus type 23 vector based vaccine efficiently protects common marmosets against Zika virus infection", has been editorially accepted for publication at PLOS Neglected Tropical Diseases.

Before your manuscript can be formally accepted and sent to production you will need to complete our formatting changes, which you will receive in a follow up email. Please note: your manuscript will not be scheduled for publication until you have made the required changes.

IMPORTANT NOTES

* Copyediting and Author Proofs: To ensure prompt publication, your manuscript will NOT be subject to detailed copyediting and you will NOT receive a typeset proof for review. The corresponding author will have one final opportunity to correct any errors when sent the requests mentioned above. Please review this version of your manuscript for any errors.

* If you or your institution will be preparing press materials for this manuscript, please inform our press team in advance at plosntds@plos.org. If you need to know your paper's publication date for media purposes, you must coordinate with our press team, and your manuscript will remain under a strict press embargo until the publication date and time. PLOS NTDs may choose to issue a press release for your article. If there is anything that the journal should know, please get in touch.

*Now that your manuscript has been provisionally accepted, please log into EM and update your profile. Go to http://www.editorialmanager.com/pntd, log in, and click on the "Update My Information" link at the top of the page. Please update your user information to ensure an efficient production and billing process.

*Note to LaTeX users only - Our staff will ask you to upload a TEX file in addition to the PDF before the paper can be sent to typesetting, so please carefully review our Latex Guidelines [http://www.plosntds.org/static/latexGuidelines.action] in the meantime.

Best regards,

David W.C. Beasley

Associate Editor

Sunit Singh

Deputy Editor

---

## [Editor Report · Acceptance letter]

27 Jan 2020

Dear Prof. Li,

We are delighted to inform you that your manuscript, "A high infectious simian adenovirus type 23 vector based vaccine efficiently protects common marmosets against Zika virus infection," has been formally accepted for publication in PLOS Neglected Tropical Diseases.

Best regards,

Serap Aksoy

Editor-in-Chief

Shaden Kamhawi

Editor-in-Chief
